# Analysis of arsenic-modulated expression of hypothalamic estrogen receptor, thyroid receptor, and peroxisome proliferator-activated receptor gamma mRNA and simultaneous mitochondrial morphology and respiration rates in the mouse

Daiana Alymbaeva[1], Csaba Szabo[2], Gergely Jocsak [1]*, Tibor Bartha[1], Attila Zsarnovszky[1,2,3], Csaba Kovago[4], Silvia Ondrasovicova[5], David Sandor Kiss[1]

1 Department of Physiology and Biochemistry, University of Veterinary Medicine, Budapest, Hungary,
2 Department of Animal Physiology and Health, Hungarian University of Agricultural and Life Sciences, Godollo, Hungary, 3 Agribiotechnology and Precision Breeding for Food Security National Laboratory, Department of Animal Physiology and Health, Institute of Physiology and Nutrition, Hungarian University of Agricultural and Life Sciences, Kaposvar, Hungary, 4 Department of Pharmacology and Toxicology, University of Veterinary Medicine, Budapest, Hungary, 5 Department of Biology and Physiology, University of Veterinary Medicine and Pharmacy in Košice, Košice, Slovakia

* jocsak.gergely@univet.hu

## Abstract

Arsenic has been identified as an environmental toxicant acting through various mechanisms, including the disruption of endocrine pathways. The present study assessed the ability of a single intraperitoneal injection of arsenic, to modify the mRNA expression levels of estrogen- and thyroid hormone receptors (ERα,β; TRα,β) and peroxisome proliferator-activated receptor gamma (PPARγ) in hypothalamic tissue homogenates of prepubertal mice *in vivo*. Mitochondrial respiration (MRR) was also measured, and the corresponding mitochondrial ultrastructure was analyzed. Results show that ERα,β, and TRα expression was significantly increased by arsenic, in all concentrations examined. In contrast, TRβ and PPARγ remained unaffected after arsenic injection. Arsenic-induced dose-dependent changes in state 4 mitochondrial respiration (St4). Mitochondrial morphology was affected by arsenic in that the 5 mg dose increased the size but decreased the number of mitochondria in agouti-related protein- (AgRP), while increasing the size without affecting the number of mitochondria in pro-opiomelanocortin (POMC) neurons. Arsenic also increased the size of the mitochondrial matrix per host mitochondrion. Complex analysis of dose-dependent response patterns between receptor mRNA, mitochondrial morphology, and mitochondrial respiration in the neuroendocrine hypothalamus suggests that instant arsenic effects on receptor mRNAs may not be directly reflected in St3-4 values, however, mitochondrial dynamics is affected, which predicts more pronounced effects in hypothalamus-regulated homeostatic processes after long-term arsenic exposure.

**Data Availability Statement:** All relevant data are within the paper and its Supporting Information files.

**Funding:** This work was funded by the Hungarian Scientific Research Fund, OTKA K-115613 (to Attila Zsarnovszky; https://www.otka-palyazat.hu/) and was partly funded by the Ministry of Human Resources, EFOP-3.6.3-VEKOP-16-2017-00005 (to Attila Zsarnovszky; https://univet.hu/hu/egyetem/palyazati-projektek/eu-projektek/tudomanyos-utanpotlas-erositese/efop-3-6-3-vekop-16-2017-00005-mentorprogram/). This work was also supported by the Agribiotechnology and Precision Breeding for Food Security National Laboratory, Institute of Physiology and Nutrition, Department of Animal Physiology and Health, Hungarian University of Agriculture and Life Sciences, 7400 Kaposvár, Hungary) no. RRF-2.3.1-21-2022-00007 (to Attila Zsarnovszky; https://nkfih.gov.hu/national-laboratories). The sponsors or funders have not been involved in the study design, data collection and analysis, decision to publish, or preparation of the manuscript

**Competing interests:** The authors have declared that no competing interests exist.

# 1. Introduction

Based on the latest analysis conducted by the Agency for Toxic Substances and Disease Registry (ATSDR), and the Environmental Protection Agency, which accounts for factors such as substance toxicity, potential human exposure, and frequency of occurrence, arsenic (As) has been identified as the most hazardous substance. This recent (2022) ranking is documented in their priority list, accessible at https://www.atsdr.cdc.gov/spl/index.html#2019spl. The issue of arsenic poisoning has become increasingly prevalent in many developing nations, often stemming from contamination of drinking water sources due to natural environmental factors, such as seepage into aquifers, mining activities, and industrial operations [1]. Globally, more than 150 million people are impacted by arsenic-contaminated drinking water, posing a substantial public health concern due to the chronic nature of exposure [2]. Both human population studies and animal research have consistently demonstrated that chronic exposure to As leads to cognitive impairment [3, 4].

Many of the environmental toxicants act as agonists or antagonists of certain hormones (mainly estrogens and/or thyroid hormones [5–11]); interfering with the physiological regulation of the homeostatic system. These chemicals have been termed endocrine disruptors (EDs) [12].

In general, within the frame of ED-modulated intracellular hormone actions, there are two major and fundamentally interrelated types of intracellular mechanisms. №.1.: regulatory pathways that regulate physiological balance, optimizing responses to the environment [13–16]; №.2.: adjustment of cellular energy expenditure to homeostatic processes [17–19].

Emerging evidence suggests that As exposure disrupts the normal functioning of the endocrine system, influencing some intricate regulatory processes related to hormone synthesis and secretion [20]. Given their integral roles in orchestrating diverse physiological processes, the thorough investigation of the thyroid and estrogen systems is of fundamental importance in the assessment of a substance demonstrating potential endocrine-disrupting properties [21–23]. Apart from these "traditional" hormone receptors, some studies highlight that the adipogenesis activators like peroxisome proliferator-activated receptor gamma (PPARγ), a crucial regulator of energy metabolism [24, 25], is also sensitive to As exposure [26] that raises the question of whether PPARγ is also a potential target for endocrine disruptors [27].

While the toxicological impact of As is widely acknowledged in various aspects, the mechanisms through which As elicits its endocrine disrupting effects on the above hormonal systems, remain largely unknown. Nevertheless, our earlier studies have already demonstrated some aspects of its disrupting potential in *in vitro* conditions [28–31] however, until now it remained questionable whether these phenomena may also manifest *in vivo*.

Located in the basal region of the brain, the hypothalamus serves as a key regulator of the above-mentioned hormonal systems, functioning as an intricate network comprising specialized neurosecretory cells [32]. Receiving a myriad of external and internal signals through hypothalamus-pituitary-end-organ axes [33, 34], these cells compose the major regulatory center for homeostatic processes [35]. Among the numerous hypothalamic nuclei that integrate the processes related to energy balance, nutrient intake, and feeding behavior, the arcuate nucleus (ARC) receives special attention in our present research since it is positioned at a region with higher local permeability in the blood-brain barrier, making it readily exposed to circulating factors, such as chemicals expressing endocrine disrupting capability [36–39]. ARC neurons, particularly those expressing proopiomelanocortin (POMC; termed as "satiety neurons"), and agouti-related peptide (AgRP; so-called "hunger cells") not only play a vital role in sensing the overall energy status of the organism as key members of the melanocortin system but also integrate signals from both central and peripheral pathways [40–42].

Mitochondria in somatic cells typically serve general cellular metabolic functions, producing ATP, while–quite interestingly–mitochondria located in AgRP and POMC neurons are suggested to play a specific role in regulating the energy balance of the whole organism through dynamic changes in their activity that are believed to directly mimic and modulate their hosting cells' function; i.e. they act as energy sensors of the hosting neurons [43–45]. Regarding the context between hormone receptors and cellular energy management, the influence of ER, TR, and PPAR on mitochondria is unequivocal, manifesting through both direct and indirect pathways across diverse cellular contexts [46–49].

These effects encompass interactions with nuclear and extranuclear receptors, as well as non-genomic interactions with other organelles and intracellular events. Given the heightened presence of ER and TR within the hypothalamus, coupled with their capacity to traverse the blood-brain barrier and undergo local synthesis within the brain, it is crucial to comprehend the individual and collective impact of ER and TR on mitochondrial function [50, 51]. Considering the distinctive roles played by AgRP- and POMC-associated mitochondria and their profound impact on the overall organism, it becomes apparent that EDs acting upon these hormone receptor pathways possess the capacity to modulate an extensive array of systemic mechanisms. This modulation may result in altered physiological functions or even contribute to the onset of different diseases.

We hypothesize that As can act as an endocrine disruptor in low doses and can also *in vivo* modulate the expression of different hormone receptors (ERα,β, TRα,β and PPARγ) within the hypothalamus. Coupled with that, different intracellular pathways are suggested to be affected leading to altered function of organelles, most prominently those including mitochondria, which–in the case of AgRP and POMC cells–are remarkably sensitive to circulating factors, like hormone-mimicking chemicals. Therefore, beyond the above we investigated the mitochondrial respiration rates (MRR), a generally accepted parameter for the assessment of the intensity of mitochondrial metabolism, moreover, we examined the mitochondrial morphology in AgRP and POMC cells to offer insights into the broader mechanisms governing metabolic regulation.

In determining our arsenic dosage, we referenced several key studies. The selection of the quantities of hormones and EDs, specifically 40 µg, 5 mg, and 10 mg, was based on several factors. Firstly, these dosages were chosen according to the sensitivity of the applied neuronal cell culture system and our previous experiments where the most effective dosage in proving ED effects was determined [52, 53]. These quantities represent varying levels of arsenic exposure, reflecting different degrees of environmentally relevant concentrations of 40 µg, 5 mg, and 10 mg were chosen to represent varying exposure levels, with 40 µg falling within WHO limits (permissible limit of 10–50 µg) [54]. Chang et al. (2007) [55] noted reproductive effects in mice at 20–40 mg/l sodium arsenite exposure. Jana et al. (2006) [56] found neuroendocrine effects in rats at 5 mg/kg/day sodium arsenite. Moreover, Stump et al. (1999) [57] administered intraperitoneal doses (0, 5, 10, 20, 35 mg/kg) during in-utero development, observing the effects of As exposure during critical developmental periods. On the other hand, the concentrations of 5 mg and 10 mg, as cited in studies by Smedley and Kinniburgh, (2012) [58] and Shankar, (2014) [59], respectively, demonstrate higher exposure levels that exceed permissible limits. Our approach to determining intervention time in our animal model was based on insights from our previous comprehensive pilot study [28]. Initially, we conducted *in vitro* experiments assessing TR and ER receptor expressions with their ligands using cerebellar granule cells. We maintained a consistent intervention time of 6 hours for PCR analysis to ensure reliable results, aligning with the peak expression period observed, we also noted a secondary peak for Western blot analysis around 18 hours. Subsequently, we refined these time windows through further *in vitro* experiments examining hormone receptor expressions post-

exposure to endocrine disruptors [30, 31, 60]. By systematically progressing through these stages and aligning intervention times across *in vitro* and *in vivo* experiments, we aimed to ensure consistency and reliability in our findings while effectively investigating the effects of arsenic exposure on hormone receptor expression in the hypothalamus.

We believe that exposure to elevated levels of As may exert modulatory effects on homeostatic functions by disrupting the integrity of the hypothalamic melanocortin system. This perturbation has the potential to induce enduring physiological consequences, including but not limited to conditions such as diabetes, accelerated aging, and cognitive impairments (Fig 1).

## 2. Materials and methods

### 2.1. Animals and treatments

18-day-old C57BL/6 mice were used for this study, (purchased from HAS Biological Research Centre, Szeged, Hungary). Since pilot studies showed no differences between the examined parameters in the two sexes, and we found no indication in the relevant literature for hypothalamic gender differences in this respect, pups of both sexes were used (weighing 9–10 g, n = 6 per treatment group, separately for PCR, MRR and electron microscopic stereological measurements). The age chosen represents a pre-pubertal state when the hypothalamus is not yet sexually active, however, being just before the onset of reproductive life, it is responsive to sexual steroids. This period marks a crucial transitional phase, highlighted by Sengupta P. et al., (2017 a,b) [61, 62] and Sengupta T. et al., (2021) [63], and Brydges N. (2016) [64]. Rebuli and Patisaul, (2016) [65] also emphasized the significant impacts of EDs on the hypothalamus during this developmental stage. While behavioral effects of EDs have been extensively studied, neural changes in pre-pubertal animals have received comparatively less attention.

Animals were kept under a 12/12-h light/dark cycle illumination program. Animals were fed with regular chow (vendor: FarmerMix Kft., Zsambek, Hungary) and ad libitum tap water. All experimental procedures were conducted at the University of Veterinary Medicine (Budapest, Hungary) in accordance to ARRIVE guidelines and EU Directive 2010/63/EU, and was reviewed and approved by the Animal Health and Animal Welfare Directorate of the National Food Chain Safety Office (permit no.: XIV-I-001/2202-4/2012, PEI/001/665-8/2015 and PE/EA/1252-6/2016), as well conformed to ARRIVE guidelines and EU Directive 2010/63/EU.

Animals were sorted into experimental groups according to doses administered as mentioned below, plus a control group injected with the vehicle only (non-treated control, ntC). Hypothalamic samples were measured for the following parameters: mRNA expression level of ERα, ERβ, TRα TRβ, PPARγ.

Six hours before sampling, animals received a single intraperitoneal injection of sodium (meta)arsenite (arsenic, As; purity: ≥90%; CAS: 7784-46-5), purchased from Sigma-Aldrich (St. Louis, MO, USA). Injections were given in three different doses: 40 μg, 5 mg and 10 mg As per 1000 g body weight, dissolved in 0.9% NaCl + dimethyl sulfoxide (0.1%) and DMSO solution. Non-treated controls (ntC) were injected with the solvent as vehicle only.

### 2.2. Sampling

**2.2.1. Sampling for tissue homogenates.** Sampling was performed through quick guillotine decapitation under deep isoflurane narcosis 6 hours after the treatments, as described earlier [28, 29, 53]. Removal of the hypothalamic tissue followed the anatomical protocol previously described [66]. Briefly, hypothalami were dissected from the removed brains as follows: in anterio-posterior direction: between the caudal margin of the optic chiasm and the rostral margin of the mamillary body and in dorsoventral direction: below the upper margin of the fornix. Both sides of the hypothalami were used for preparing the tissue homogenates

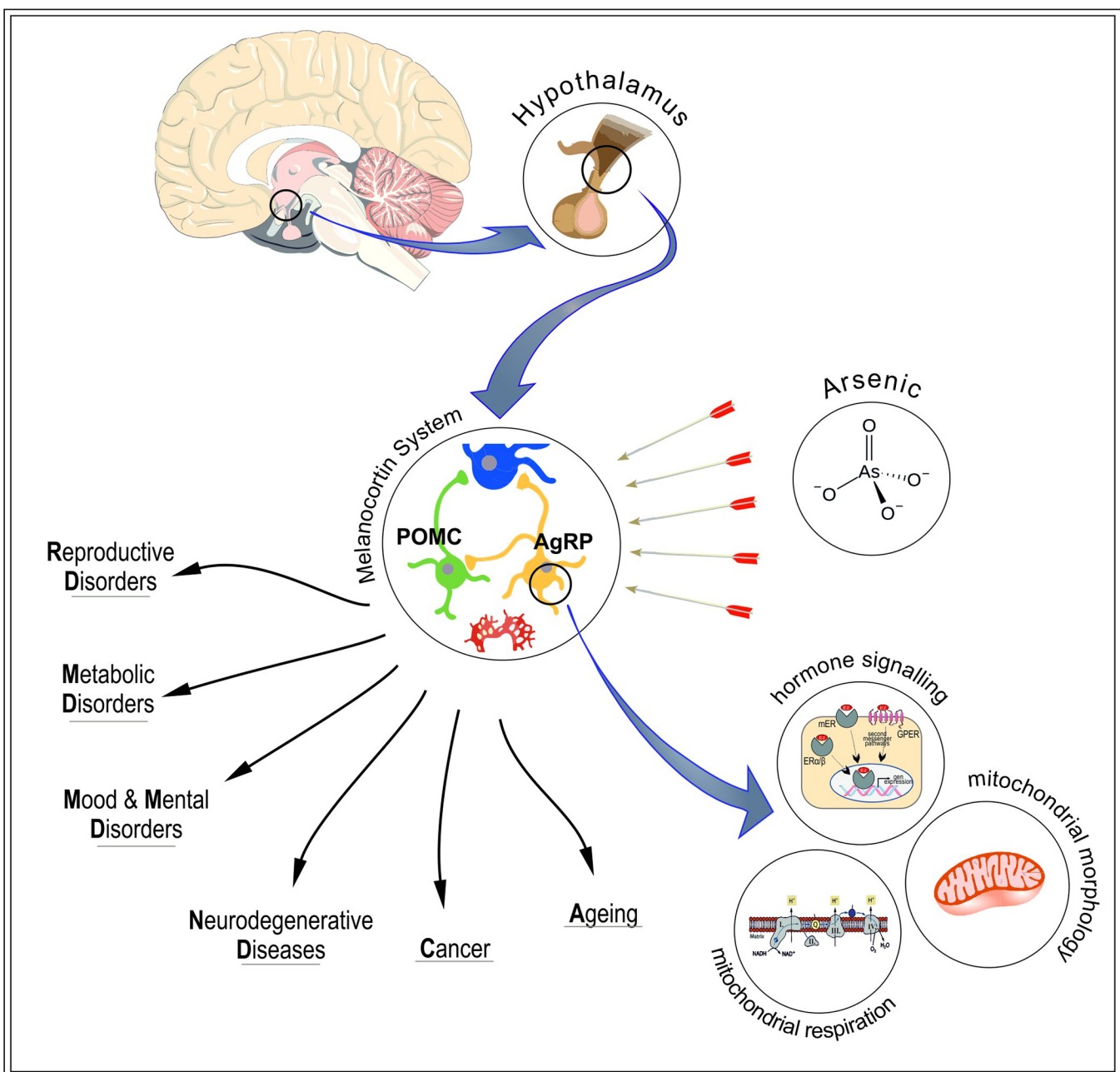

**Fig 1. The hypothesized mechanisms underlying arsenic-induced endocrine disruption, emphasizing its potential impact on the primary endocrine center.** The melanocortin system within the hypothalamus emerges as a key target of arsenic, with adverse effects anticipated across diverse cellular processes, including mitochondrial respiration and morphology, as well as hormonal signaling pathways. Persistent dysfunction of the melanocortin system is implicated in the etiology of various systemic disorders, such as metabolic syndromes, reproductive dysfunction, mood disorders, and neurodegenerative diseases.

(although we have taken measures to develop a suitable method for the measurement of mitochondrial respiration rates, in relatively small samples [67], still, using only half of the mouse hypothalamus provided a sample size too small to reliably obtain MRR data). Samples were then used for the determination of the level of receptor mRNAs and mitochondrial respiration rates, as detailed below.

**2.2.2. Sampling for electron microscopy.** Anesthetized animals (details above) were perfused through the left cardiac ventricle, first with 50 ml of 0.9% NaCl, followed by 250 ml of 4% paraformaldehyde and 0.5% glutaraldehyde in 0.1 M phosphate buffer (PB), pH 7.4. After perfusion, the isolated brains were placed in 4°C glutaraldehyde-free fixative (4% paraformaldehyde) for an additional 3 h. Samples were prepared for immunohistochemistry (IHC) by cutting the isolated brain blocks into 50-μm-thick sections (this thickness ensures optimal tissue morphology preservation and staining efficiency. Specifically, when employing 4% paraformaldehyde fixation, sections within the 20–50 μm range facilitate efficient antibody penetration and antigen retrieval).

## 2.3. Quantitative-RT-PCR

Total RNAs were isolated from mouse hypothalami using TRI reagent following the manufacturer's protocol (Invitrogen, Carlsbad, CA, USA, #AM9738) and purified from samples with the Direct-zol RNA miniprep kit (Zymo Research, Irvine, CA, USA, #R2051). RNA levels were determined spectrophotometrically (NanoDrop™ ND-1000, Wilmington, NC, USA) at 260–280 nm absorption. 3 μg/μL of total RNA was reverse transcribed by RT-PCR (Amplitron II., Barnstead/Thermolyne, Dubuque, IA, USA) using M-MLV reverse transcriptase (Promega Corporation, Wisconsin, USA, #M1701), oligo (dt) primers and dNTPmix.

Subsequently, 2 μL of the resulting cDNA samples were analyzed by qRT-PCR (Master SYBRGreen, F. Hoffmann-La Roche, Basel, Switzerland) in triplicates, in a LightCycler 2.0 device (F. Hoffmann-La Roche, Basel, Switzerland). To evaluate qPCR reactions, constitutively expressed genes β-actin (ACTB), beta-glucuronidase (GUSB), glyceraldehyde 3-phosphate dehydrogenase (GAPDH), β-2-microglobulin (B2M) and transferrin receptor protein 1 (TFRC) were evaluated for stability. Transcription levels of the genes under investigation were normalized using the expression values of GAPDH, which was identified as the most consistently expressed gene across the experimental conditions, as determined by the GeNorm and NormFinder algorithms. These algorithms were accessed through the online platform RefFinder [68]. Primer pairs were designed using NCBI's primer designer Primer-BLAST, or were taken from literature, and used at 2 μM concentration. The primer sequences used for GAPDH, ERα, ERβ, TRα, TRβ and PPARγ are given in Jocsak et al., 2019 [29], (please see Table 1). qPCR cycles and controls were planned according to the manufacturer's instructions and were optimized for the primer pair as described by Jocsak et al., 2016 [31]. Real-time PCR threshold cycle (Ct) data were analyzed using the REST-XL software version 2.0 (GenEx-

**Table 1. Primer sequences used for qRT-PCR analysis.**

| Target gene (mouse) | Primer Sequence 5'–3' | |
|---|---|---|
| ERα | Forw. | GGA ACT GTC TGC CCA TCG TT |
| | Rev. | GAA CCC AGG GCT GCC TTA C |
| ERβ | Forw. | AAC CTT CCT CTT GGG CAT CG |
| | Rev. | TTT CAT CCG GTT CTC CCA CC |
| TRα | Forw. | ACC GCA AAC ACA ACAT TCC G |
| | Rev. | GGG CCA GCC TCA GCT AAT AA |
| TRβ | Forw. | CGA GGC CAG CTG AAA AAT GG |
| | Rev. | CTC AGC ACA CTC ACC TGA AGA |
| PPARγ | Forw. | TTGGTGGGATTGTGTCTCGG |
| | Rev. | GGCCAAGATCTCACAGTGCT |
| GAPDH | Forw. | TGA AAT GTG CAC GCA CCA AG |
| | Rev. | GGG AAG CAG CAT TCA GGT CT |

BioMcc, TUM, München, Germany) [69]. Cycle threshold values were normalized to those of GAPDH. The relative expression ratios of mRNA (fold changes) were calculated using the $2^{-\Delta\Delta Ct}$ method.

## 2.4. Mitochondria metabolism

**2.4.1. Isolation of mitochondria.** Hypothalamic samples were prepared for measurement of oxygen consumption as described by Kiss et al., 2016 [69]. Briefly, unit amounts of hypothalamic tissue homogenates (in isolation buffer) were used for mitochondrial purification by a Percoll gradient fractionation. Mitochondrial oxygen-consumption was measured by a Clark-type oxygen electrode (Oxytherm, Hansatech Instruments, Norfolk, UK) at 37°C. Measured values represent the mitochondrial respiration rate (MRR, given in consumed nmol $O_2$ per ml of final volume in one minute). Mitochondrial sample homogenates included both the left and right sides of the hypothalami (i.e., the left and right sides were not separately processed and measured).

**2.4.2. Definition of mitochondrial respiration states.** As the name and numeral marking of different mitochondrial respiration states varies in the relevant literature, subsequently we explain our nomenclature as used in the present study. Explanation of mitochondrial respiration states as sequentially measured (60 seconds for each respiration state (please also see: [66]).

First step: the mitochondrial oxygen consumption was measured in respiration buffer only, without the addition of any substrates that may affect mitochondrial respiration. Under such conditions, oxygen consumption per unit time depends on the actual metabolic state of the hypothalamic sample and the sample's original oxygen supply. We termed this experimental setup as state 1 mitochondrial respiration (St1).

Second step: to fuel the Krebs cycle, 5 μL pyruvate (P; comprising the following mixture: 275 mg pyruvate/5 mL distilled water + 100 μL 1 mol/L HEPES) and 2.5 μL malate (M; comprised the following mixture: 335.25 mg malate/5 mL distilled water +100 μL 1 mol/L HEPES) were added to the sample. Under such conditions, the Krebs cycle intensifies, and oxygen consumption increases due to consequential facilitation of the terminal oxidation and oxidative phosphorylation, if the prior (*in vivo*) blood/oxygen supply of the hypothalamic tissue was sufficient and downregulating mechanisms were not active. We termed this experimental setup as state 2 mitochondrial respiration (St2).

Third step: adenosine diphosphate (ADP; comprising of the following mixture: 64.1 mg ADP/5 mL distilled water + 100 μL 1 mol/L HEPES) of 2.5 μL was added to the sample. Since ADP is a major upregulator of mitochondrial respiration, under such conditions MRR increases if the prior (*in vivo*) blood/fuel supply of the hypothalamic tissue was sufficient. We termed this experimental setup as (ADP-dependent) state 3 mitochondrial respiration (St3).

Fourth step: oligomycin (comprised of the following mixture: 1 mg oligomycin/1 mL ethanol) of 1 μL was added to the sample. Oligomycin is an ATP-synthase blocker, therefore, it inhibits the oxidative phosphorylation (ATP synthesis), while terminal oxidation continues. Under such conditions, oxygen consumption depends on the actual uncoupled stage and alternative oxidation in mitochondria. Uncoupling and alternative oxidation play important roles in transient downregulation of ATP biosynthesis when cellular energy needs drop. Therefore, increased oxygen consumption in this case refers to the decline of a process (that was previously upregulated) or the attempt by the mitochondrion to downregulate ATP synthesis. We termed this experimental setup as state 4 mitochondrial respiration (St4).

Fifth step: carbonylcyanide-4-(trifluoromethoxy)-phenyl-hydrazone (FCCP; comprised the following mixture: 1.271 mg FCCP/5 mL dimethyl sulfoxide) of 2.5 μL was added to the sample. The FCCP is a cyanide derivative; therefore, by binding to, and blocking cytochrome C

oxidase, it depletes all remaining oxygen from the sample. A decrease in oxygen level under such conditions depends on the actual/initial (*in vivo*) metabolic state of the sampled tissue and the amount of oxygen consumed during St1-4. Thus, the total amount of oxygen consumed in St1-4 plus the amount of remaining oxygen depleted by FCCP gives good reference to the blood/oxygen supply of the tissue at the time of the animal's sacrifice. Therefore, this experimental setup is also known as total mitochondrial respiratory capacity, hereby referred to as state 5 mitochondrial respiration (St5).

Although all 5 mitochondrial respiration states (as explained above) were sequentially measured and evaluated, only MRR from state 3 (St3) and state 4 (St4) mitochondrial respiration values are presented in this study, since these MRR parameters are most appropriate to characterize the intensity of mitochondrial/tissue metabolism. St3 gives a plausible insight into mitochondrial metabolism, since the ADP/ATP ratio potently regulates mitochondrial activity, while St4 indicates the degree of uncoupling and the activity of alternative oxidases, two factors that play an important role in transient down-regulation of ATP biosynthesis when cellular energy needs drop.

### 2.5. Immunocytochemistry & electron microscopy

50 μm thick tissue slices [70] were rinsed in 1% sodium borohydride in PB for 15 min to eliminate unbound aldehydes. The primary antibodies used for labelling included a polyclonal primary POMC (Invitrogen PA5-18368, 1:1500, incubated overnight) and AgRP antibodies (Invitrogen PA5-78739, 1:200, incubated overnight). As a secondary antibody a biotinylated polyclonal goat anti-rabbit antibody was applied (Invitrogen 65–6140, 1:200, incubated for 1 hr.). Finally, ABC kits (avidin-biotin complex kits, # Pk.4001) and DAB (diaminobenzidine, CAS: 91-95-2) were used for development.

For electron microscopic analysis, sections were osmicated (1% $OsO_4$ in PB) for 30 min, dehydrated through increasing ethanol concentrations, and embedded into Durcupan ACM resin (Sigma-Aldrich, 44610). Following embedding, ultrathin sections with a thickness of 50 nm were cut for precise examination, (see above the localization of the area in question).

The prepared sections were examined with a JEOL-1011 TEM (JEOL USA, Peabody, MA, USA). The calibrated electron micrographs were analyzed using the NIH Image J (ver. 1.52n) software. The measured parameters included the occurrence (number per area) and size of mitochondria, matrix/entire mitochondrion ratio [71].

### 2.6. Data analysis

In this study, statistical analysis was conducted using GraphPad Prism version 9 (GraphPad Software, San Diego, CA, USA) and carried out according to the guidance of the Department of Biomathematics, University of Veterinary Medicine, Budapest, Hungary.

Prior to analysis, the normality of the data distribution was assessed using the Shapiro-Wilk test. As data followed a normal distribution in the data sets, one-way ANOVA was performed followed by Bonferroni or Dunnett's test for post hoc comparisons, for PCR, and MRR and for structure measurements, respectively. Additionally, the homogeneity of variances was examined using Levene's test.

Results are presented as mean ± standard deviation (SD). All statistical tests were two-tailed, and p-values less than 0.05 were considered statistically significant.

## 3. Results

In the present study, we examined the modulatory effects of three environmentally relevant doses of As on mRNA expression levels of ERα,β, TRα,β, and PPARγ, respectively;

simultaneous MRR in mouse hypothalamic tissue homogenates and mitochondrial morphology (number and size of mitochondria, and the proportional mitochondrial matrix area to the area of the entire mitochondrial crossection) were also determined. Concerning the MRR, although measurements have been carried out according to our description in the Materials and Methods section (i.e., all St1-5 were recorded), here we only present the results of St3 and St4 measurements, since these two parameters are directly relevant to demonstrate the intensity of mitochondrial metabolism (and №.2 reactions).

### 3.1. Receptor mRNA levels

In general, As treatment evoked elevated receptor expression in ERs and TRs, and even reached significance depending on the receptor subtypes. In the case of both ERα and TRα (Fig 2A and 2C.), As caused a remarkable dose-dependent increase in the receptor expressions (i.e., higher dose led to higher expression), where the treatment groups showed significant difference compared to the non-treated control and to each other, too. In contrast to TRα (Fig 2A), expression of TRβ (Fig 2B) was not affected by the As injection, since treated groups remained in the expression range observed in the case of the control. Also, the ERβ expression levels (Fig 2C) were drastically changed in a dose-dependent manner as a result of As-treatment, however, the pattern of the different doses was also different from that of ERα (Fig 2D): the expression in all treatment groups increased significantly compared to the control and each other. However, the highest peak was observed in the intermediate dose. Similarly to the TRβ groups, As only negligibly altered the PPARγ expression, without regard to the treatment doses (Fig 2E).

### 3.2. Mitochondrial respiration St3-4

Arsenic induced dose-dependent changes in MRR in both St3 and St4 values, their pattern showing a broad similarity to each other; however, that of the St4 was stronger (Fig 2F). Administration of the highest As dose (10 mg) decreased the mitochondrial activity (significantly in the case of St4), while the further two doses showed roughly the same activity as observed in the control. It is also important to note that St4 MRR values demonstrate a clear (and significant) inverse concentration dependence, i.e. as the As dose was reduced, the MRR increased to near the control mean (please see 40 μg group).

### 3.3. Mitochondrial ultrastructure

In both the AgRP (Fig 3A, 3C, 3E, 3G) and the POMC (Fig 3B, 3D, 3F, 3H) neurons, the size of mitochondria increased exclusively in animals treated with 5 mg of As ($p < 0.001$ and $p < 0.05$, respectively), while in the 40 μg and 10 mg group no significant change could been observed compared to the control (Fig 3A and 3B.).

The number (per 10 $\mu m^2$ section area) of mitochondria has changed only in the 5 mg group, demonstrating a slight decrease ($p < 0.1$) in AgRP neurons (Fig 3C).

Proportional mitochondrial matrix area (matrix area per entire mitochondrion area in electron microscopic section) showed a drastic increase ($p < 0.001$) in a dose-dependent manner (higher As concentration caused higher density) (Fig 3E and 3F).

We also examined the juxtaposition of mitochondria and the endoplasmic reticulum in both the AgRP-IR (Fig 3G) and POMC-IR neurons (Fig 3H), however, neither the average distance between the two organelles, nor the number of mitochondrion-endoplasmic reticulum contacts were changed in any of the experimental groups.

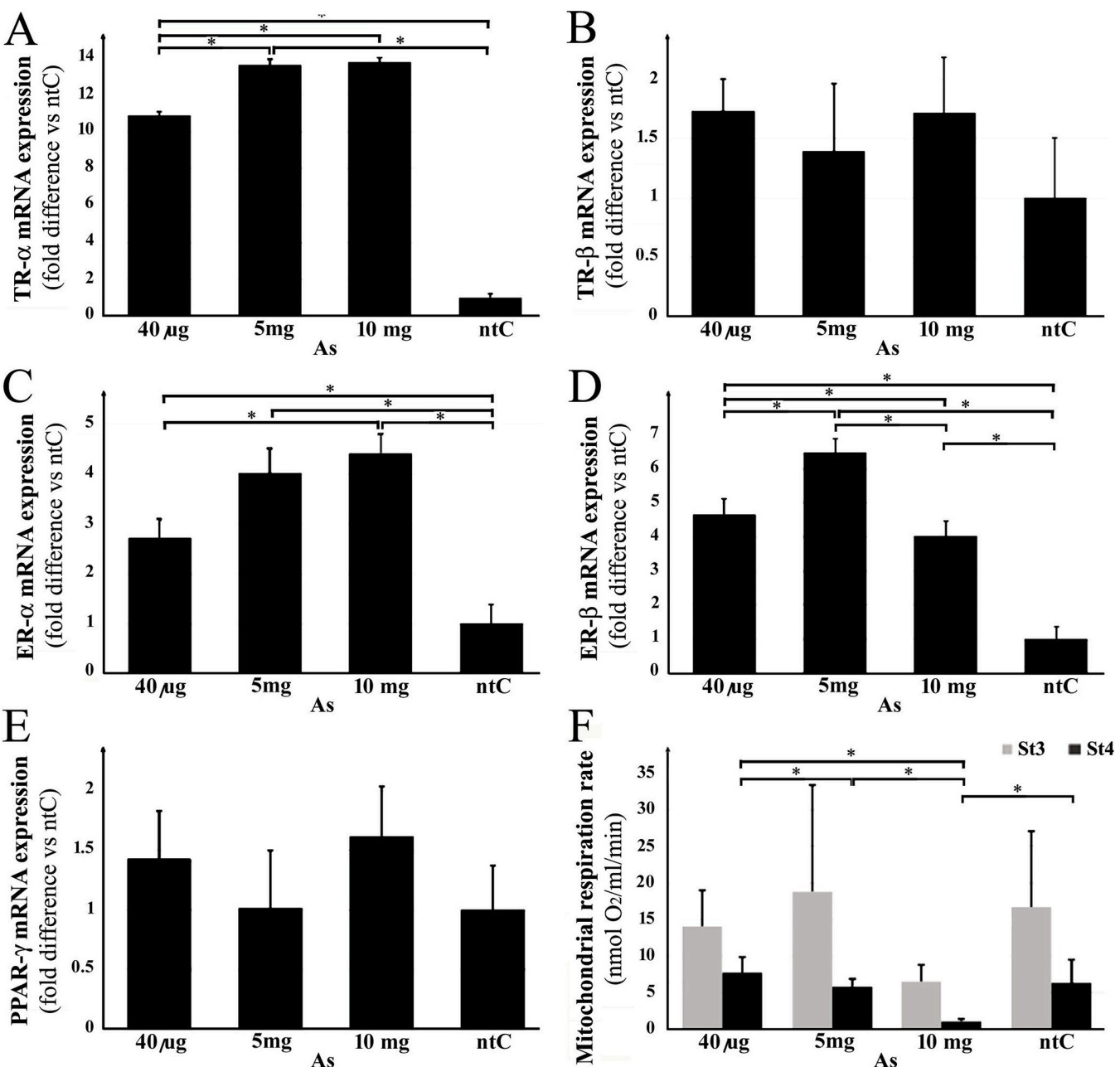

**Fig 2. Effects of 3 different doses of a single intraperitoneal injection of sodium (meta)arsenite (arsenic, As) on the expression of hypothalamic nuclear hormone receptors (ERα,β; TRα,β, PPARγ), and hypothalamic mitochondrial state 3–4 respiration.** (A-E): Relative expression level of receptor genes was analyzed by qRT-PCR and normalized to the average of the control gene GAPDH. (**F**): ADP-dependent (St3) and oligomycin-induced (St4) mitochondrial respiration rates were given in consumed nmol $O_2$ per ml of final volume in one minute. The data shown here are the mean ± standard deviation (SD) and compared to the non-treated control (ntC). *P*-values are represented by asterisk (*) where $p < 0.05$.

## 3.4. Behavior of experimental animals

Mice that received the highest dose (10 mg) showed loss of physical activity, depression-like behavior, ataxia, loss of interest for feed and mild trembling 1.5–3 hours after receiving the intraperitoneal injection of As (similar administration of other EDs did not evoke such behavioral responses [unpublished observation]). Autopsy of these animals revealed no macroscopic pathological symptoms.

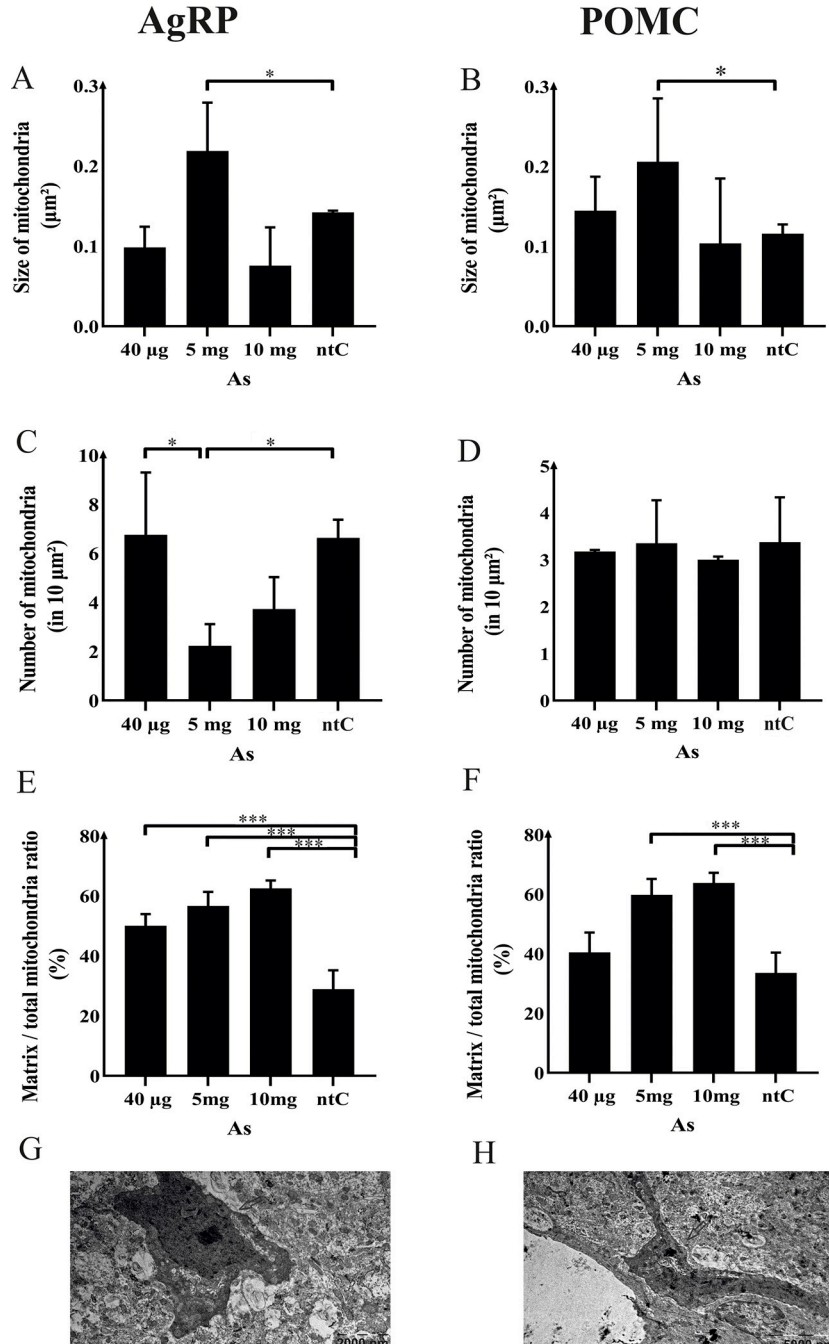

**Fig 3. The effects of 3 different doses of a single intraperitoneal injection of sodium (meta)arsenite (As) on the mitochondrial microstructure in AgRP and POMC neurons.** (A-B) the size of mitochondria, (C-D) the number of mitochondria is measured per unit section area. (E-F) the proportional mitochondrial matrix (matrix-to-total mitochondrial ratio) area is expressed as a percentage. (G-H) the images of electron micrographs of AgRP and POMC neurons, where arrows indicate examples of POMC-IR and AgRP-IR neurons. The micrographs showcase the effectiveness of the staining technique employed in this study, providing high-quality visualization of the cellular structures of interest. The data presented above (A-F) are expressed as the mean ± standard deviation (SD) and compared to the non-treated control (ntC) group. $P$-values are represented by asterisks (*,**) where $p < 0.05$ and $p < 0.001$, respectively.

## 4. Discussion

### 4.1. Receptor mRNA levels

The literature abundantly reports how As exposure is associated with different endocrine disorders; however, a mechanistic investigation is still lacking [72, 73]. Arsenics's impact on the hypothalamic-pituitary-ovarian axis is already unquestionable. Nevertheless, the impact of As on the estrogen signalization remains, at least in some senses, inconsistent, since As can either trigger or inhibit ER expression and ER-related pathways, with or without regard to its doses, tissues and species drawn into investigation [74–79]. In our case, As treatment generally resulted in an elevated expression in both ER receptors. For example, ERβ expression levels (Fig 2D) were drastically changed in a dose-dependent manner, however, this change was different from that observed regarding ERα expression (Fig 2C). It is to note here, that ERα is recognized as the primary mediator of estrogenic effects on energy balance and an essential factor in the regulation of energy homeostasis, is mainly found in brain regions associated with the control of energy balance more precisely in the melanocortin system [80]. Our present findings are in accord with some previous literature data, however, our assessment of acute As exposure adds new findings to the existing knowledge from long-term exposure studies. Non-cytotoxic doses of As have already proved to strongly suppress ER-dependent gene transcription in *in vivo* poultry and *in vitro* human models [81]. This suggests that As may bind to ERs rendering them inaccessible for their natural ligands, a mechanism that can lead to an upregulation of ER expression in trying to regain the native functions. Regardless of whether As increases or decreases receptor expression levels, it is evident that ER-mediated №.1 intracellular processes are affected. Since ERs are not only present in the host cell but also in the mitochondria, it is likely that №.2 reactions are also promptly affected by AS exposure. This idea could be paralleled with our finding that As likewise seemed to depress dose-dependently St4: considering the interpretation of St values described in the *Materials and Methods* section, a decrease in St4 is a sign of an attempt of the mitochondria to produce more ATP. In contrast to this, Chen et al. [82] detected significant inhibition of ERα expression after long-term exposure to low doses of $As_2O_3$, but no effects have been detected with regard to ERβ. Another long-term study also reported the downregulation of ERα along with the decrease of the estrogen-responsive gene activities after a 4 µg/ml exposure to As [79]. Thus, our present results suggest that the immediate effects of As on ERs expression might differ from the long-term effects.

Studies on the relation of thyroid receptors and exposure to As are scarcer than those on estrogen-driven mechanisms. The majority of the reported studies examined thyroid markers other than the relevant receptors (in human patients), however, only a single article discussed the receptor expression (in amphibians). They reported that even very low As level can profoundly affect TR-dependent developmental processes in vertebrates [83]. According to some other studies, sub-chronic As exposure can significantly reduce TRβ expression in the rodent cerebellum both on mRNA and protein levels [84]. In humans, As treatment resulted in an elevated expression of TR mRNA, and also TSH was positively associated with the plasma As [85], however, the data on thyroid hormones (T3, T4) are again controversial [86, 87]. Compared to these, in our experiments, TRα mRNA was, significantly increased, regardless of the doses that we applied (Fig 2A); meanwhile, however, TRβ (Fig 2B) was not affected by the As-injection. The observed differences between As effects on TRs may lie on the grounds of several reasons: besides interspecies differences, direct hormonal and As environment of cells in question may significantly differ; and it is highly likely that significant differences between immediate and chronic effects of exposure also exist.

In this study, As altered only negligibly the PPARγ expression, with no regard to the injected doses (Fig 2E). In contrast, the literature reports variegated outcomes. Arsenic (3.80 ppm) increased PPARγ expression levels, followed by apoptosis, in *in vitro* cultured rat astrocytes [88]. PPARγ has also been reported to mediate the impact of As on increasing the incidence of type 2 diabetes through the PPARγ-mTORC2 signaling pathway [89, 90]. In preadipocytes, arsenic trioxide inhibited cell differentiation into fat cells by suppressing the PPARγ expression [91–93]. In our present study, we detected no immediate effect of As exposure on PPARγ expression. However, this finding does not mean that PPARγ is insensitive to all EDs: our simultaneous experiments show that some other EDs like bisphenol A and zearalenone, in low concentrations, are able to increase PPARγ expression under similar experimental conditions that we applied in the present study (unpublished observations).

## 4.2. Mitochondrial respiration St3-4

Arsenic neurotoxicity and the adverse (usually drastic inhibitory) effects of As on brain mitochondria are already well-established. Toxic As doses, as a result of chronic exposure, induce super-oxide anion and ROS formation [83, 94, 95], decrease the activity of mitochondrial complexes I, II and IV [96], inhibition of ATPase activity [96–98], dissipated mitochondrial transmembrane potentials [99–102], as well as affect the mitochondrial proton motive force [95, 97, 103], moreover, leads to malformation of mitochondria [83, 104]. However, it is important to note that our current understanding does not include an examination of low (or non-toxic) doses of As in relation to mitochondrial function. Despite this, our results align with existing literature as we observed a significant inhibition of mitochondrial uncoupling activity at the highest dose of As used. In contrast, the lower two doses showed minimal deviation from the control group, demonstrating an inverse concentration-dependence. This pattern resembles the effects of certain E2 agonists [105–107] (see Fig 2C and 2D). Interestingly, the effects of long-term exposure (days to weeks [102, 104]) to As, i.e., inhibition of ATP synthesis by mitochondria is preceded by an increase in ATP production as indicated by the observed low levels of St4 in the 10 mg group. With this regard, an additional important finding demonstrates that alterations in the uncoupling potency of the mitochondria in the melanocortin system are implicated in the intracellular processes that determine the output signals of the AgRP and POMC cells [108].

If this is the case, it suggests that 10 mg of As instantly propels mitochondrial ATP production, followed by the exhaustion of the mitochondria. This observation adds important information to our knowledge of As effects when discussing these effects in the context of №.1 and №.2 reactions. As mentioned in the case of the nuclear receptor expression, the finding that As only affected MRR in its St4 and only in 10 mg dose does not mean that mitochondrial metabolism is not sensitive to other EDs: various doses of bisphenol A and zearalenone had profound effects on St3-4 under experimental conditions that we applied for As exposure (unpublished observations). Those studies will further broaden our view of the temporal relationship of №.1 and №.2 reactions in response to various ED toxicants.

## 4.3. Mitochondrial ultrastructure

When discussing the ultrastructural features of mitochondria, we took into consideration the generally accepted interpretation of their morphological parameters. Accordingly, increased size, as well as decreased overall numbers of mitochondria (in AgRP and POMC cells) suggest organelle fusion. Increased fusion and/or decreased fission helps overcome low levels of stress (e.g., starvation [109, 110] or toxic effects [111]). Decreased fusion and/or increased fission occurs with high levels of stress (e.g., during apoptosis [112, 113] Increase in the size of the

mitochondrial matrix per the entire mitochondrial crossection area is commonly associated with mitochondrial swelling, specifically in the context of oncotic or apoptotic mitochondria. This event is accompanied by a concomitant suppression of oxidative phosphorylation and a consequent decrease in ATP production [114]. More precisely, dynamic changes of mitochondria located in the melanocortin system are suggested to mimic the energy status of the organism (i.e. acting as energy sensors) and are actively participating in the modulation of the cellular activity (neurotransmission and release of cell-specific peptides) of their hosts, the AgRP and POMC neurons. Originating from their mitochondrial responses, these neurons are subsequently engaged in orchestrating distinctive processes governing energy expenditure across the organism and conveying pertinent information to higher brain centers [108].

In our present study As caused the increase of mitochondrial size at 5 mg dose only. With the interpretation described above in mind, this finding suggests that the aforementioned dose of As imposed a mild stress on mitochondria; however, definitions like those above, as taken from the literature, do not count with the versatility in the multitude of parameters of the cellular environment and the length of exposure. Thus, if we want to hold to generally accepted ways of interpretation, we have to emphasize the likely temporal changes in ED effects as cells try to acclimate to the various concentrations of EDs in the short- and long run. This means that, in our case, 5 mg of As may instantly cause a compensating adaptive response in mitochondria, but we cannot rule out that later the same dose could result in the exhaustion of mitochondria with inversed intensity in its metabolic activity [26, 115, 116]. At the same time, doses other than 5 mg may also be considered effective when examined after long-term exposure, although the present study does not answer this question.

The number (per section area) of mitochondria has changed only in the 5 mg group, demonstrating a slight decrease ($p < 0.1$) in AgRP neurons. It is not easy to interpret the meaning of this observation, considering that the number of mitochondria may increase as a physiological need for more ATP production (e.g., mitochondrial division in hypothalamic neurons after the mid-cycle estrogen surge [43, 108]) but it may also increase as a result of fission, latter which is the result of various stress effects and carries morphological signs such as mitochondrial swelling. With this in mind, our finding that the proportional size of the mitochondrial matrix (indicative of the level of mitochondrial swelling) increased linearly with the injected doses of As, suggests that the increase in the size of mitochondria with a simultaneous decrease in their numbers clearly shows immediate stressful and deleterious effect of As on AgRP neurons. The latter phenomenon may be an adaptive process aimed at mitigating the deleterious effects that pose a threat to the survival of the cell by compromising its metabolic and structural integrity. Beyond this, a decreased number of mitochondria paralleled with higher size of them, is characteristic of AgRP cells in the case of high-fat diet conditions [108]. This further supports the idea, that As may modulate the regulation of the energy homeostasis through affecting melanocortin neurons.

Arsenic's immediate effect on POMC cells, at the same time, did not affect the number of mitochondria; this, however, does not exclude the possibility of effects on morphology after longer As exposure. The latter, seemingly speculative statement is supported by our findings that mitochondrial swelling occurred in a dose-linked manner (Fig 3A). It seems to be evident that As can alter the ultrastructure of mitochondria in hypothalamic neurons; however, here we only examined two distinct types of versatile neuron populations of the hypothalamus. Therefore, results from our studies do not mean that similar results occur in hypothalamic neurons other than AgRP and POMC cells; on the other hand, considering the rich neuronal interconnectivity between the hypothalamic neurons, it is highly likely that functional alterations in neuronal functions in response to As exposure may also play a role in the appearance

of mitochondrial ultrastructure be it in the cell types examined in the present study or other cells of the hypothalamus.

Finally, the effects of arsenic, as observed and described so far, raise the question of how mitochondrial functions and related cellular metabolism in the hypothalamus could be affected after such changes. Firstly, animals showed signs of anxiety -/depression-like behavior. Numerous reports describe such effects of arsenic [117–119], although those effects were observed after a longer time of exposure. These findings indicate the complexity of As's effects, in that the aforementioned behavioral symptoms are regulated by a wide variety of brain regions, among them the loss of interest in feed-uptake involves the hypothalamus and its "hunger cells" (AgRP neurons) and "satiety cells" (POMC neurons). At the same time, however, among the ultrastructural results, the changes in size and number of mitochondria were observed, nevertheless, these morphological alterations were found in the 5 mg group instead of the 10 mg group. In the 10 mg group, as shown by the dose-dependent values of matrix-entire mitochondrion ratio (Fig 3E and 3F), the most severe changes appeared in the form of alterations of the behavior, as described above (3.4. of the Results section). Thus, it seems that initial changes in the number and size of mitochondria can be interpreted as compensational reactions to less toxic doses of As, whereas mitochondrial swelling (also known as the initial sign towards mitochondrial apoptosis) occurs at higher doses of As and such a toxic dose also manifests in the form of altered behavior. The finding that no changes occurred in the juxtaposition of mitochondria and endoplasmic reticulum suggests that immediate effects of As do not include mass growth in the volume of interaction between these organelles, namely, toxically affected mitochondria are likely to enter apoptosis [120–123] instead of a generalized mitochondrial compensatory action in cellular metabolism.

## 4. Conclusion

The study confirms our hypothesis that even low doses of arsenic prompt immediate biological effects within hours of exposure. These effects involve rapid alterations in major hormone systems, particularly estrogen receptor (ER) and thyroid receptor (TR) expression, suggesting arsenic's role as an endocrine disruptor targeting the neuroendocrine center. Considering these, arsenic influences energy expenditure by modulating cellular metabolism, appetite, and thermogenesis through hypothalamic estrogen and thyroid signaling pathways. Arsenic also affects mitochondrial dynamics in the hypothalamic melanocortin system, resembling the effects of a chronic high-fat diet.

Altogether, arsenic's significant influence extends to the delicate and bidirectional interaction between the periphery and the central nervous system governed by the hypothalamus, potentially leading to endocrine dysfunction and the onset of metabolic diseases over time. These effects involve №.1 (intracellular regulatory pathways) and №.2 (adjustments of cellular energy expenditure) mechanisms, highlighting the complexity of arsenic's impact on cellular and systemic processes.

The findings underscore the importance of further exploring the dose- and time-dependent effects of arsenic alone and in combination with other endocrine disruptors in the environment, to better understand their implications for human health and well-being.

## Supporting information

**S1 Raw data. It was obtained from electron microscopy (JEOL USA, Peabody, MA, USA).** (XLSX)

**S2 Raw data. It was measured by a Clark-type oxygen electrode (Oxytherm, Hansatech Instruments, Norfolk, UK).**
(XLSX)

**S3 Raw data. It obtained from PCR.**
(XLSX)

## Author Contributions

**Conceptualization:** Attila Zsarnovszky, David Sandor Kiss.

**Data curation:** Daiana Alymbaeva, Gergely Jocsak.

**Funding acquisition:** Tibor Bartha, Attila Zsarnovszky.

**Investigation:** Daiana Alymbaeva, Gergely Jocsak.

**Methodology:** Attila Zsarnovszky, David Sandor Kiss.

**Resources:** Csaba Szabo, Tibor Bartha.

**Supervision:** Csaba Szabo, Attila Zsarnovszky, David Sandor Kiss.

**Validation:** Attila Zsarnovszky.

**Visualization:** Gergely Jocsak, Attila Zsarnovszky.

**Writing – original draft:** Gergely Jocsak, Attila Zsarnovszky.

**Writing – review & editing:** Csaba Kovago, Silvia Ondrasovicova, David Sandor Kiss.

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
