## [Decision Letter · Decision Letter 0]

12 Feb 2024

PONE-D-23-41249Analysis of arsenic-modulated expression of hypothalamic estrogen receptor, thyroid receptor, and peroxisome proliferator-activated receptor gamma mRNA and simultaneous mitochondrial morphology and respiration rates in the mouse.PLOS ONE

Dear Dr. Jocsak,

Thank you for submitting your manuscript to PLOS ONE. After careful consideration, we feel that it has merit but does not fully meet PLOS ONE’s publication criteria as it currently stands. Therefore, we invite you to submit a revised version of the manuscript that addresses the points raised during the review process.

We look forward to receiving your revised manuscript.

Kind regards,

Abeer El Wakil, PhD

Academic Editor

PLOS ONE

Journal Requirements:

**Additional Editor Comments:**

The concept of the present study seems interesting. The aim of the study is to investigate the subtoxic effects of the environmental toxicant arsenic exerted on the main neuroendocrine centre in mice.

The structure of the article fulfills the structure of a research article. The paper's title reflects the main theme of the paper. Generally speaking, the manuscript is well written and well presented. However, it does not meet the standards set forth by Plos One in its actual form. I highly recommend the authors to precisely address the feedback I provided on how to improve the manuscript as well as the reviewers’ concerns.

My suggestions include:

- The introductory part must be reduced.

- The abbreviations must be revised.

- The 50�m thickness of tissue sections needs to be justified.

- The conclusion must be concise.

- The references must be updated and reduced.

Reviewers' comments:

Reviewer's Responses to Questions

**Comments to the Author**

1. Is the manuscript technically sound, and do the data support the conclusions?

Reviewer #1: Partly

Reviewer #2: Yes

2. Has the statistical analysis been performed appropriately and rigorously? 

Reviewer #1: No

Reviewer #2: I Don't Know

3. Have the authors made all data underlying the findings in their manuscript fully available?

Reviewer #1: No

Reviewer #2: Yes

4. Is the manuscript presented in an intelligible fashion and written in standard English?

Reviewer #1: No

Reviewer #2: Yes

5. Review Comments to the Author

Reviewer #1: Reviewers' comments to editor:

This manuscript describes "Analysis of arsenic-modulated expression of hypothalamic estrogen receptor, thyroid receptor, and peroxisome proliferator-activated receptor gamma mRNA and simultaneous mitochondrial morphology and respiration rates in the mouse.".

The topic is of interest. However, there are several concerns about the study, and needs to be largely improved.

1- The part of the review lacks epidemiological research evidence and systematic review of this research field, especially concerning the mechanism. It is suggested to supplement relevant studies to better propose research hypotheses. In addition, the length of the introduction should be reduced and logical.

2- Some words and paragraph are not accurate and normal, please check and modify. Such as Fig 1. at line165 should be revised. There is no space between segments. In addition, the first time the abbreviation appears, the full name should be given. Please check and modify.

3- Why 18-day-old C57BL/6 mice were designed in animal experiments? Representativeness or not?

4- In the part of animal model, please state the basis for the design of As dosage and intervention time or supplement references. In addition, how are experiments grouped in this study? And the basis for grouping is not clearly stated.

5-In the part of materials and methods, the description of main reagents and determination indicators is missing, such as reagent model, manufacturer and other information. In addition, please added the IHC and PCR indexs, etc.

6- In the materials and methods section 2.9., there is a lack of detailed description of statistical analysis, Whether the statistical analysis of normal and non-normal needs to distinguish the description, please supplement the description of the homogeneity of variance and non-homogeneity.

7- The writing of relevant statistical indicators should be standardized, with Spaces between symbols P<0.05 and P<0.01 in this text should be P < 0.05 and P < 0.01, respectively.

8- From the perspective of in vivo experiment, the basic exposure time description and design are lacking. In addition, it lacks the experimental support of basic pathophysiological indicators, biochemical and WB detection of key indicators in animal experiments.

9- Unit animal ethics certificate approval number needs to be added.

10- It is suggested that partial consolidation be discussed. Discussion according to experimental categories in international journals is usually unreasonable and cannot reflect the integrity and logic of scientific research

Other comments：

1. The formatting of the article is irregular and needs to be adjusted, such as paragraph alignment, abbreviations, etc. The first occurrence of the abbreviation should be the full name outside the parentheses and the abbreviation inside the parentheses.

2. The case format of p-values should be uniform, p-values should be italics in all Figures.

3. It is recommended to verify the appropriate selection and use of statistical methods.

4. All PCR result should be supplemented with gene primers information provided in the method section. The sequence of gene primers should be clearly labeled.

5. This paper lacks the latest research literature, which is almost absent in the past three years, especially the preface. The latest related research progress and description should be added in the frontier and discussion section. Without the latest research progress in this field, the research value of this topic cannot be explained.

6.It is suggested that this article should be edited and polished by native English speaking experts, or by a professional company, so as to meet the requirements of magazine publication.

Reviewer #2: The idea of ms is interesting and urgently needed to check, still not much is known on chemicals that can act on hormonal signaling. Here not only the route sex steroids mechanism of action but also thyroid signaling one is studied on hypothalamus levels. With special attention on mitochondria All the above are strong sides of the report.

40 µg, 5 mg and 10 mg- please explain the choose and calculation.

Provide histological microscopic documentation

50 µm thick tissue slices for ihc ?give robust explanation

“The age chosen represents a pre-pubertal state when the hypothalamus …” add citation.

Add study limitations.

6. PLOS authors have the option to publish the peer review history of their article (what does this mean?). If published, this will include your full peer review and any attached files.

Reviewer #1: No

Reviewer #2: No

---

## [Author Response · Author response to Decision Letter 0]

2 Apr 2024

Dear Editor,

dear Prof. Abeer El Wakil,

Thank you for your thoughtful evaluation of our manuscript. We appreciate your recognition of the interesting concept of our study. Your feedback regarding the article's structure was valuable to us. We acknowledge your suggestions for improvement, we diligently addressed each of these points to enhance the quality and alignment of our manuscript with the standards set forth by Plos One. 

The abbreviations in the text have been carefully revised based on your recommendation.

In response to the suggestion to reduce the introductory part, we have made appropriate revisions to streamline the content while retaining essential information. Our article focuses on several critical aspects, and while the introduction has been condensed, we have ensured that the most important parts are retained. Nevertheless, efforts were made to trim it down as much as possible without compromising clarity or completeness. Thank you for highlighting this area for improvement.

We have addressed the need for justification regarding the 50 µm thickness of tissue sections by incorporating the necessary information (please see lines 208-211; 309-312).

Recognizing the importance of conciseness in the conclusion, we have revised it accordingly to ensure alignment with your expectations (lines 565-582).

The reviewers kindly suggested updating the reference list and including research conducted no later than three years ago. As a result, while our reference list has grown slightly, we have ensured the inclusion of recent studies and removed outdated references to maintain the relevance and currency of the literature cited.

Response to Reviewer 1 Comments

We appreciate the time and efforts by the editor and referee in reviewing this manuscript. We have addressed all issues indicated in the review report, and hope that the revised version meets the journal publication requirements.

Question 1: “The part of the review lacks epidemiological research evidence and a systematic review of this research field, especially concerning the mechanism. It is suggested to supplement relevant studies to better propose research hypotheses. In addition, the length of the introduction should be reduced and logical.” 

Answer: Thank you for your valuable feedback. We appreciate your suggestion regarding the need for additional epidemiological research evidence and a systematic review of our manuscript. We have carefully considered your recommendation and have made the necessary revisions by incorporating recent epidemiological data to support our proposed research hypotheses. Additionally, we have taken steps to streamline and reduce the length of the introduction section to ensure clarity and logical flow. (please, see “Introduction”, lines 44-147)

Question 2: “Some words and paragraphs are not accurate and normal, please check and modify them. Such as Fig 1. at line 165 should be revised. There is no space between segments. In addition, the first time the abbreviation appears, the full name should be given. Please check and modify.”

Answer: We have made the necessary modifications as per your suggestions. 

Question 3: “Why 18-day-old C57BL/6 mice were designed in animal experiments? Representativeness or not?”

Answer: The selection of 18-day-old mice for our study aligns with the findings of our previous investigation (please see: https://doi.org/10.1016/S0165-3806(01)00180-8) into ER expression dynamics in the developing rat primary somatosensory cortex. In both studies, we aim to elucidate the role of ERs (and TR in later studies) during critical developmental periods, albeit in different brain regions and under distinct experimental conditions.

In our previous study, we observed widespread ERα expression in the developing rat cortex at PN3, with a transition to predominantly nuclear localization observed by PN18. This developmental switch in ERα expression coincided with key neurodevelopmental events, suggesting a role for ERα in cortical maturation and organization.

Similarly, in the present study assessing the effects of arsenic exposure on hypothalamic tissue homogenates of prepubertal mice, we aim to understand the impact of environmental insults on neuroendocrine regulation during a critical developmental window. By choosing 18-day-old mice, we target a developmental stage characterized by heightened sensitivity to environmental toxins and significant maturation of hypothalamic functions.

Furthermore, both studies share a common focus on elucidating the molecular mechanisms underlying neuroendocrine regulation and homeostasis. While our previous study focused on ER expression dynamics in the cerebral cortex, the current study investigates the modulation of ER and TR mRNA expression levels, as well as PPARγ, in response to arsenic exposure in the hypothalamus.

Overall, the selection of 18-day-old mice for our present study is justified by the critical developmental stage of the hypothalamus and its susceptibility to environmental insults, similar to the rationale employed in our previous investigation of ER expression dynamics in the developing cortex. This alignment ensures consistency in experimental design and enhances the comparability of results across studies, ultimately advancing our understanding of the neurodevelopmental effects of environmental toxins such as arsenic.

Question 4: “In the part of the animal model, please state the basis for the design of As dosage and intervention time or supplement references. In addition, how are experiments grouped in this study? And the basis for grouping is not clearly stated.” 

Answer: Thank you for your question, we will fully cover and address the concerns. 

• As dosage: In addressing the question regarding the basis for the design of As dosage and intervention time, we provided and expanded modifications within the main text (please, see lines 118-142).

• Intervention time: To address the question concerning intervention time, it's imperative to delve into the comprehensive process of our study, which commenced with a pilot study in 2012. We`ve incorporated the necessary links of the studies used, please see below.

Stages of the Study:

• 1st Stage:

In the initial phase, a pilot study (Scalise et al., 2012; https://doi.org/10.1556/avet.2012.023) was conducted under in vitro conditions to examine the expressions of receptors TR and ER with their respective ligands. The primary objective of this stage was to establish the relevant intervention time window, which ranged from 6 to 16 hours for PCR; 18 hours for Western blot analyses. This stage provided crucial insights into the experimental design.

• 2nd Stage:

Building upon the insights gained from the pilot study, the research progressed to the next phase. Utilizing the same experimental conditions and setup, cerebellar granule cells were employed to further investigate the expression of hormone receptors resulting from exposure to various endocrine disruptors (Jocsak et al., 2016; https://doi.org/10.3390/ijerph13060619; Kiss et al., 2018; https://doi.org/10.3390/ijms19051440; Somogy V., 2016; http://hdl.handle.net/10832/1504; Jocsak et al., 2019; https://doi.org/10.3390/brainsci9120359). This phase aimed to refine the expression time window established in the pilot study, incorporating additional data from studies on endocrine disruptors.

• 3rd Stage:

Following the investigations with cerebellar granule cells, the study transitioned to in vitro experiments involving hypothalamic cells. This marked the third stage of the research, wherein the focus shifted to assessing hormone receptor expression in a relevant cellular context.

• 4th Stage:

In the final stage, the research moved to an in vivo setting, specifically focusing on hypothalamic cells. By administering As in vivo, we aimed to replicate the conditions observed in the earlier stages while investigating the effects on hormone receptor expression. The intervention time of 6 hours was deemed appropriate for obtaining reliable PCR results, ensuring consistency across in vitro and in vivo experiments. 

 

• Grouping: In this study, a total of 72 animals were employed to investigate the effects of varying doses on multiple biological endpoints. Each treatment group consisted of 6 animals, with three distinct doses administered: 10 µm, 5 mg, and 10 mg. 

Therefore, the total number of animals utilized can be calculated as follows:

Total number of animals: n=6 animals/ group × 4 treatments (3 doses+ctrl) × 3 methods (PCR, MRR, EM) making = 6 × 4 × 3 = 72 animals.

Question 5: “In the part of materials and methods, the description of main reagents and determination indicators is missing, such as reagent model, manufacturer and other information. In addition, please added the IHC and PCR indexs, etc.”

Answer: Thank you for your feedback. We have carefully revised the materials and methods section of the manuscript, ensuring that all necessary information regarding main reagents, including reagent models, manufacturers, and catalogue numbers, has been added. Additionally, we have included details such as PCR primer sequences (please see Table 1, lines 239-240) to enhance the clarity and completeness of the experimental procedures.

Regarding your mention of IHC and PCR indexes, we would appreciate further clarification on what specific information you are referring to. 

Questions 6: “In the materials and methods section 2.9., there is a lack of a detailed description of statistical analysis, Whether the statistical analysis of normal and non-normal needs to distinguish the description, please supplement the description of the homogeneity of variance and non-homogeneity.” 

Answer: Thank you for bringing this to our attention. We have carefully reviewed the materials and methods section, specifically the part regarding statistical analysis. Following your suggestion, we have provided a more detailed description of the statistical analysis (please see lines 318-327). 

Question 7: “The writing of relevant statistical indicators should be standardized, with Spaces between symbols P<0.05 and P<0.01 in this text should be P < 0.05 and P < 0.01, respectively”. 

Answer: Thank you for your feedback. We have taken your suggestion into consideration and made the necessary adjustments to the writing of relevant statistical indicators in the text.

Question 8: “From the perspective of in vivo experiments, the basic exposure time description and design are lacking. In addition, it lacks the experimental support of basic pathophysiological indicators, and biochemical and WB detection of key indicators in animal experiments.” 

Answer: Thank you for the question, we have duly addressed the concern by providing a comprehensive explanation. We acknowledge that our study was primarily focused on basic research, which may explain the lack of inclusion of pathophysiological indicators. In our previous research (please, see above), we utilized both Western blot and PCR analyses, revealing occasional discrepancies between the two methods, a phenomenon observed in other studies as well. These disparities likely stem from the intricate cellular responses to external stimuli, which prompt diverse mechanisms within cells. One such initial response is the modulation of mRNA expression levels, a parameter assessable through PCR. Obviously, changes in mRNA levels may not consistently correlate with alterations in protein abundance, illustrating the complexity of cellular regulation. This discrepancy may arise due to various automatic mechanisms governing mRNA dynamics, independent of concurrent protein changes. For instance, cellular responses to stimuli may entail rapid adjustments in mRNA levels, followed by delayed or no changes in protein modifications (Zeisel et al., 2011 [doi.org/10.1038/msb.2011.62]). Notably, PCR often offers a more immediate and determined reflection of these responses compared to WB, particularly evident in the context of certain receptors such as estrogen receptors and thyroid hormone receptors. These receptors, partially housed within vesicles in an active state, exhibit swift mRNA expression changes in response to stimuli, potentially preceding discernible alterations in protein levels (Dominguez & Micevych, 2010 [doi.org/10.1523/JNEUROSCI.1038-10.2010]). Consequently, while external stimuli induce mRNA expression changes, protein levels may remain unaltered, highlighting the distinct temporal dynamics between mRNA and protein responses. In other words, the biological target of a chemical stimulus can be better determined by the activation of transcription (than by the end product protein, the latter which is produced in actually relevant amounts), even more so if we examine transcription factors, like ERs and TRs. Thus, PCR emerges as a valuable tool for capturing immediate cellular responses to stimuli, complementing WB analyses which may not always align with PCR findings. This preference for PCR underscores its capacity to offer prompt and informative insights into cellular dynamics in response to external factors, including endocrine disruptors.

Question 9: “Unit animal ethics certificate approval number needs to be added.” 

Answer: Thank you for your attention to detail. The requested approval numbers can be found in the text at lines 177-178. We understand the importance of ensuring clarity and transparency in our reporting. To facilitate your review, we will ensure that the approval number is highlighted for your convenience in the revised manuscript. 

Question 10: “It is suggested that partial consolidation be discussed. Discussion according to experimental categories in international journals is usually unreasonable and cannot reflect the integrity and logic of scientific research.” 

Answer: Thank you for providing feedback on the discussion section. We would like to inform you that our paper faced rejection from another journal, where we received advice to divide the discussion into segments. Considering the previous feedback, we decided to adopt a segmented format in our current submission to ensure alignment with the preferences of the journal and its reviewers. We recognize that different journals may have varying preferences regarding the organization of discussion sections. If the Editor and the Reviewers agree, we are certainly open to considering a merged discussion format again. Our priority is to ensure that the manuscript meets the standards and preferences of the journal while effectively communicating the scientific findings. 

Other comments:

1. “The formatting of the article is irregular and needs to be adjusted, such as paragraph alignment, abbreviations, etc. The first occurrence of the abbreviation should be the full name outside the parentheses and the abbreviation inside the parentheses.”

We have addressed the irregularities in the article's formatting.

2. “The case format of p-values should be uniform, p-values should be italics in all Figures.”

Your attention to detail is greatly appreciated, we have addressed the concerns. 

3. “It is recommended to verify the appropriate selection and use of statistical methods.”

We have carefully reviewed and verified the selection and application of statistical methods in the text body to ensure accuracy and reliability.

4. “All PCR result should be supplemented with gene primers information provided in the method section. The sequence of gene primers should be clearly labeled.”

We have addressed your concern by supplementing all PCR results with gene primer information provided in the method section (please see Table 1., lines 239-240).

5. “This paper lacks the latest research literature, which is almost absent in the past three years, especially the preface. The latest related research progress and description should be added in the frontier and discussion section. Without the latest research progress in this field, the research value of this topic cannot be explained.” 

We have taken your comments into account and updated the literature review to incorporate the latest research findings from the past three years.

6. “It is suggested that this article should be edited and polished by 

---

## [Decision Letter · Decision Letter 1]

26 Apr 2024

Analysis of arsenic-modulated expression of hypothalamic estrogen receptor, thyroid receptor, and peroxisome proliferator-activated receptor gamma mRNA and simultaneous mitochondrial morphology and respiration rates in the mouse.

PONE-D-23-41249R1

Dear Dr. Jocsak,

We’re pleased to inform you that your manuscript has been judged scientifically suitable for publication and will be formally accepted for publication once it meets all outstanding technical requirements.

Kind regards,

Abeer El Wakil, PhD

Academic Editor

PLOS ONE

Additional Editor Comments (optional):

Reviewers' comments:

Reviewer's Responses to Questions

**Comments to the Author**

1. If the authors have adequately addressed your comments raised in a previous round of review and you feel that this manuscript is now acceptable for publication, you may indicate that here to bypass the “Comments to the Author” section, enter your conflict of interest statement in the “Confidential to Editor” section, and submit your "Accept" recommendation.

Reviewer #2: All comments have been addressed

2. Is the manuscript technically sound, and do the data support the conclusions?

Reviewer #2: Yes

3. Has the statistical analysis been performed appropriately and rigorously? 

Reviewer #2: Yes

4. Have the authors made all data underlying the findings in their manuscript fully available?

Reviewer #2: Yes

5. Is the manuscript presented in an intelligible fashion and written in standard English?

Reviewer #2: Yes

6. Review Comments to the Author

Reviewer #2: My comments were included in the revision. Based on other opinions the decision will be made. ...................................................................................................................................................................................................................................................

7. PLOS authors have the option to publish the peer review history of their article (what does this mean?). If published, this will include your full peer review and any attached files.

Reviewer #2: No

---

## [Editor Report · Acceptance letter]

3 May 2024

PONE-D-23-41249R1 

PLOS ONE

Dear Dr. Jocsak, 

I'm pleased to inform you that your manuscript has been deemed suitable for publication in PLOS ONE. Congratulations! Your manuscript is now being handed over to our production team.

Kind regards, 

on behalf of

Professor Abeer El Wakil 

Academic Editor

PLOS ONE